# Short-Term Evaluation of Cellular Fate in an Ovine Bone Formation Model

**DOI:** 10.3390/cells10071776

**Published:** 2021-07-14

**Authors:** Hareklea Markides, Nicola C. Foster, Jane S. McLaren, Timothy Hopkins, Cameron Black, Richard O. C. Oreffo, Brigitte E. Scammell, Iria Echevarria, Lisa J. White, Alicia J. El Haj

**Affiliations:** 1Guy Hilton Research Centre, Institute of Science and Technology in Medicine, Keele University, Thornburrow Drive, Stoke-on-Trent ST4 7QB, UK; Markidesh@gmail.com (H.M.); tim.hopkins102@googlemail.com (T.H.); i.echevarria@keele.ac.uk (I.E.); 2Healthcare Technologies Institute, Institute of Translational Medicine, School of Chemical Engineering, University of Birmingham, Birmingham B15 2TT, UK; N.C.Foster@bham.ac.uk; 3Centre for Biomolecular Sciences, University of Nottingham, Nottingham NG7 2RD, UK; jane.mclaren@nottingham.ac.uk (J.S.M.); lisa.white@nottingham.ac.uk (L.J.W.); 4Bone and Joint Research Group, Centre for Human Development, Stem Cells and Regeneration, Faculty of Medicine, University of Southampton, Southampton SO16 6YD, UK; cameron.black@medivet.co.uk (C.B.); Richard.Oreffo@soton.ac.uk (R.O.C.O.); 5Academic Orthopaedics, Trauma and Sports Medicine, Queen’s Medical Centre, University of Nottingham, Nottingham NG7 2UH, UK; b.scammell@nottingham.ac.uk

**Keywords:** preclinical ovine models, cell viability, magnetic nanoparticles, bone repair, mesenchymal stromal cells, osteogenesis

## Abstract

The ovine critical-sized defect model provides a robust preclinical model for testing tissue-engineered constructs for use in the treatment of non-union bone fractures and severe trauma. A critical question in cell-based therapies is understanding the optimal therapeutic cell dose. Key to defining the dose and ensuring successful outcomes is understanding the fate of implanted cells, e.g., viability, bio-distribution and exogenous infiltration post-implantation. This study evaluates such parameters in an ovine critical-sized defect model 2 and 7 days post-implantation. The fate of cell dose and behaviour post-implantation when combined with nanomedicine approaches for multi-model tracking and remote control using external magnetic fields is also addressed. Autologous STRO-4 selected mesenchymal stromal cells (MSCs) were labelled with a fluorescent lipophilic dye (CM-Dil), functionalised magnetic nanoparticles (MNPs) and delivered to the site within a naturally derived bone extracellular matrix (ECM) gel. Encapsulated cells were implanted within a critical-sized defect in an ovine medial femoral condyle and exposed to dynamic gradients of external magnetic fields for 1 h per day. Sheep were sacrificed at 2 and 7 days post-initial surgery where ECM was harvested. STRO-4-positive (STRO-4+) stromal cells expressed osteocalcin and survived within the harvested gels at day 2 and day 7 with a 50% loss at day 2 and a further 45% loss at 7 days. CD45-positive leucocytes were also observed in addition to endogenous stromal cells. No elevation in serum C-reactive protein (CRP) or non-haem iron levels was observed following implantation in groups containing MNPs with or without magnetic field gradients. The current study demonstrates how numbers of therapeutic cells reduce substantially after implantation in the repair site. Cell death is accompanied by enhanced leucocyte invasion, but not by inflammatory blood marker levels. Crucially, a proportion of implanted STRO-4+ stromal cells expressed osteocalcin, which is indicative of osteogenic differentiation. Furthermore, MNP labelling did not alter cell number or result in a further deleterious impact on stromal cells following implantation.

## 1. Introduction

The clinical adoption of regenerative cell-based therapies requires the methodical and regulated progression from bench to bedside before ultimately receiving clinical approval. Animal models are in place to not only address safety concerns, but also to assess the efficacy of the therapy prior to clinical trials [1]. The development of a bone tissue-engineered product relies on orthopaedic pre-clinical animal models of bone injury/repair implemented to test the efficacy and potency of a new therapy by the degree and quality of repair [2].

Skeletal bone defects resulting from trauma, tumour resection and disease, including osteoporosis, require immediate clinical intervention to promote good healing and restoration of function. Routine clinical protocols are successful in achieving full repair in 90% of cases within 6 months of treatment; however, in certain cases and often with no anticipation of the outcome, the defect will fail to reach a complete union, developing into clinically defined non-union bone fractures [3]. This subsequently introduces a host of challenges not only to the patient directly but also indirectly to healthcare providers and society, thus driving research towards bone tissue engineering solutions. Tissue engineering strategies typically involve the combined application of biomaterials or scaffolds to support cell delivery, provide structural support or physically stimulate bone repair. These biomaterials can be applied with or without stem/progenitor cells such as mesenchymal stromal cells and can include growth factors to trigger the complex process of repair [4].

Magnetic nanoparticle (MNP)-based technologies are new multimodal tools designed to achieve therapeutic functions in cell-based orthopaedic therapies, such as tracking, targeting and activation [5,6,7,8,9]. One such novel technology is the use of MNPs to deliver functional mechanical cues directly to cells in vivo, which has been shown to regulate cell differentiation and downstream signalling [1,10,11]. Magnetic Ion Channel Activation (MICA) defines a novel bio-magnetic approach designed to remotely deliver mechanical stimuli directly to individual cells, both in vitro and in vivo, providing a cell remote control platform, which has been shown to enhance bone formation in pre-clinical ovine models [12]. MNPs can be functionalised with biomolecules to specifically bind different cell membrane receptor targets. One such target, the mechano-sensitive ion channel TREK-1, has been shown to regulate bone stem cell differentiation into bone cell precursors in vitro [13]. Bound MNPs respond to the application of an oscillating external magnetic field, resulting in remote receptor activation and enhanced osteogenic differentiation [14,15].

The ovine critical-sized bone defect (CSBD) model provides a solid basis for testing bone tissue engineered products for use in the treatment of non-union fractures and severe trauma [16]. This is defined as the smallest defect that will not spontaneously heal during the lifetime of the animal [17,18]. The mechanisms of fracture healing are well documented in the literature but, in brief, involve the initiation of an inflammatory response with the tightly regulated interplay of multiple cell types, various cytokines, chemokines and growth factors that are all triggered within the first hours of injury and sustained for several weeks post-injury [19]. Tissue engineered products, often containing therapeutic stem and progenitor cells, would typically be exposed to this complex inflammatory environment with a limited nutrient supply, as the product is usually inserted at the time of injury [20,21].

A key question in the development of a new bone tissue engineering strategy is to define the dose of cells required within the construct. The majority of large animal bone repair studies are typically run for 3 and 12 months to evaluate medium- and long-term repair. To date, little is known of the short-term events occurring at the time of implantation. Understanding the early fate of implanted cells will help to elucidate the role of stem cell therapies and appropriate dosage to help address issues of cell viability and bio-distribution, endogenous cellular infiltration and the impact of this environment on the long-term outcomes of a study [22,23,24]. This study aims to evaluate such parameters in an ovine critical-sized defect model with and without the presence of magnetic particles and magnetic fields to understand early events potentially impacting the long-term success of the orthopaedic cell-based therapies.

## 2. Materials and Methods

Reagents were purchased from Sigma Aldrich, UK unless otherwise specified.

### 2.1. Animal Experiments

Methods were conducted as described previously [25] in accordance with the UK Home Office Regulations and protocols approved by the University of Nottingham Animal Welfare and Ethical Review Body. For all surgeries, animals were placed in lateral recumbency to allow access to the sternum and medial aspect of both hind legs. Eighteen healthy, English Mule ewes aged 2–4 years were used and assigned randomly to each treatment group. Details of experimental groups are outlined in Table 1.

Autologous mesenchymal stromal cells were isolated by bone marrow aspiration from the sternum of anaesthetised animals, as described previously [25]. The multi-lineage differentiation capacity of these cells was demonstrated in a previously published study [25]. Three weeks post-initial bone marrow harvest, a single cylindrical defect measuring 8 mm in diameter and 15 mm in depth was created in the cancellous bone region of the medial femoral condyle in the left and right hind leg of each animal. Pre-set ECM constructs containing magnetic nanoparticle-labelled and unlabelled mesenchymal stromal cells were immediately implanted within the defect using the customised delivery device, as described in a previous study [25]. Magnetic cuffs, as described previously [10], were fitted 24 h post-surgery and worn for 3 h per day until sacrifice. Sheep were sacrificed at either 2 days or 7 days post-op by an overdose of pentobarbital administered intravenously. At this point, implanted ECM constructs were harvested from the defect of sacrificed sheep, transferred directly to collection media (αMEM, 10% FBS, 1% L-glutamine and 1% AA) to maintain cell viability and transported on ice. Gel dimensions were determined by digital callipers and the presence of the implanted cells was determined by dissection fluorescent microscope. The remaining femoral condyles were then immediately trimmed and fixed in 10% neutral buffered formalin for a further 7 days prior to histological analysis.

### 2.2. Selection of STRO-4-Positive MSCs

STRO-4 selection of MSCs has previously been shown to result in enriched colony-forming fibroblasts and enhanced multi-lineage capacity in both human and ovine donors [26]. For this reason, STRO-4-positive MSCs were selected prior to expansion, as described previously [25]. In brief, the bone marrow aspirate was treated with red blood cell lysis buffer to isolate the mononuclear cell fraction. Cells were then incubated with the STRO-4 IgG hybridoma (20 µg/mL; Adelaide University) for 30 min, then with 200 µL of the magnetic-activated cell sorting (MACS) anti-mouse IgG MicroBeads (Miltenyi Biotec, Bisley, UK) (30 min, 4 °C) prior to MACS separation. Finally, STRO-4 ovine MSCs (oMSCs) were collected and plated in expansion media and maintained at 37 °C for 1 week before further media changes. Cells were cultured under standard cell culturing conditions in αMEM for on-going experiments.

### 2.3. CM-Dil Labelling

In order to track the implanted cells, they were labelled with a fluorescent dye, CM-Dil, prior to ECM encapsulation, as described previously [25]. Briefly, oMSCs were re-suspended and incubated in the working solution (3 mM) of CM-Dil for five minutes at 37 °C, and then for an additional 15 min at 4 °C, in the dark.

### 2.4. MNP Labelling of STRO-4-Positive oMSCs

oMSCs, at 80–90% confluence, were collected, counted and washed in PBS to remove any residual FBS. Cells were then re-suspended in the MNP labelling solution, which consisted of serum-free media (αMEM containing 1% L-glutamine and 1% antibiotic and anti-mycotic) and TREK-1-functionalised MNPs (Nanomag, Micromod, 1 mg/mL, Rostock, Germany). A cell labelling ratio of 25 µg MNPs per 106 cells with 1 µL liposomal transfection reagent DOTAP (1 µg/mL) was maintained throughout. Cells were labelled in suspend for 3 h at 37 °C and unbound MNPs removed by PBS wash and centrifugation (1000 rpm; 5 min). The corresponding unlabelled cell groups were simultaneously incubated in SFM only.

### 2.5. Encapsulation of oMSCs within an ECM Gel Construct for In Vivo Delivery

Preparation of the ECM digest, formed from bovine tibiae, and the resulting ECM gel (12.5 mg/mL) are described in a previously published article [27]. This concentration was previously determined to be optimal for the release of gels from the mould and their subsequent insertion into the defect site. Briefly, 5 × 10^6^ MNP-labelled or -unlabelled oMSCs from each donor were re-suspended in a 20% HEPES solution and thoroughly mixed with the ECM digest. The subsequent gel mixture was then transferred to a customised sterile delivery device complementing the dimensions of defect. The gelation occurred at 37 °C for 1 h before hydrating with SFM. Pre-set constructs were maintained at 37 °C and implanted the following day. Donor-matched in vitro controls seeded with unlabelled oMSCs were also prepared and maintained in culture for the duration of the study. SFM in control groups was changed to expansion media at the time of in vivo implantation.

### 2.6. Assessment of Cellular Viability by LDH Staining

Viability was assessed via lactate dehydrogenase (LDH) staining, as described previously [25]. Briefly, frozen sections (16 µm) were incubated with staining solution for 30 min at 37 °C. Slides were mounted with Hydromount and imaged (Nikon Eclipse, Ti-S, Minato City, Tokyo, Japan). Implanted cells were identified by red fluorescence (CM-Dil staining) and viable cells by blue staining (indicating the presence of the active LDH enzyme) under bright field settings. Ten random field of views were imaged per section in a total of five sections. Viability was evaluated with ImageJ software (Wayne Rasband, National Institutes of Health, Bathesda, MD, USA) by quantifying the proportion of dual LDH and CM-Dil staining relative to total CM-Dil staining.

### 2.7. Histology

Frozen sections were fixed in 10% formalin and stained for haematoxylin and eosin (H&E) and Prussian blue. To stain for H&E, sections were treated with haematoxylin Gill number 3 for 4 min followed by a 0.3% acid alcohol wash, rinsed in water and then briefly treated with Scott’s tap water (dH_2_O with 0.2% sodium hydrogen carbonate and 2% magnesium sulphate). Finally, an alcohol-based Eosin dye was added for 2 min and sections were immediately washed in tap water. Stained sections were mounted in an aqueous-based mounting media (Aqua-Mount) and preserved for long-term imaging. To stain and identify the presence of MNPs, a Prussian blue stain was implemented. Here, a solution consisting of 20% aqueous hydrochloric acid (HCl) and 10% aqueous potassium hexacyanoferrate was added to each section for 20 min. The presence of MNPs is identified as bright blue staining when imaged with bright field microscopy.

### 2.8. Immunocytochemistry

Sample sections (12 µm) were fixed prior to antigen retrieval (0.1% trypsin made up in 1% calcium chloride, 10 min incubation at 37 °C). Sections were subsequently blocked (3% BSA; 1 h at RT). Primary antibodies, shown in Table 2, were added and incubated at 4 °C, overnight. Upon thorough washing, secondary antibodies were added and incubated for 2 h. Finally, sections were washed, treated with DAPI to stain for nuclei and mounted with Fluoromount. Sections were imaged using the Cytation 5 imaging system. The expression of osteocalcin was quantified with ImageJ by measuring the area occupied by red fluorescence in images taken from three or four different regions of each slide. Cell number was also quantified with ImageJ using CM-Dil identify individual cells from the same regions. The osteocalcin area was then normalised to cell number.

### 2.9. Quantification of C-Reactive Protein Levels by ELISA

CRP (Neo biolabs) and non-haem iron (Randox) levels were determined by enzyme-linked immunosorbent assay (ELISA) to assess the immune response associated with MNPs and bone ECM implantation and circulating non-haem iron, which is an indication of MNP degradation. Serum was collected from each animal at sacrifice at either 2 or 7 days post-cell implantation and compared to pre-implantation levels. A total of 10 mL of blood was collected from the jugular vein in untreated 20 mL falcon tubes (no anticoagulant) from each sheep prior to cell delivery (day 0) and upon sacrifice (days 2 and 7). Serum was collected by allowing blood to coagulate overnight at 4 °C, followed by centrifugation at 2000× *g* for 30 min. The CRP ELISA was conducted according to manufacturer’s instructions.

### 2.10. Statistical Analysis

All statistical analyses were performed using GraphPad Prism version 7.00 for Windows (GraphPad Software, San Diego, CA, USA). A regular two-way ANOVA, followed by Dunnett’s multiple comparisons test with Tukey correction, was used to compare the differences in cell viability. A one-way ANOVA, followed by Dunnett’s multiple comparisons test with Tukey correction, was used to compare differences in osteocalcin expression.

## 3. Results

### 3.1. Implants Remained Intact with No Degradation Observed at Either 2 or 7 Days Post-Implantation

Animals were sacrificed at either 2 or 7 days post-implantation. The long-term fate of implanted cells was examined in a previous study [25]. Upon sacrifice, joints were re-opened for total examination and removal of the implanted construct for further examination (Figure 1). A haematoma was observed at the defect site, completely encapsulating the implanted construct in all groups and at both time points (Figure 1A). The hydrogel construct remained intact and the dimensions unchanged (6.44 ± 0.68 × 14.83 ± 1.2 mm) when compared to the initial pre-implantation in vitro control dimensions (8 × 15 mm), with no observed differences between time points (Figure 1B). Finally, whole-mount fluorescent microscopy confirmed the presence of implanted CM-Dil-labelled oMSCs, as seen by the red fluorescence in all groups (Figure 1C).

### 3.2. Significant Loss in Cell Viability Determined at 7 Days Post-Implantation

LDH is an enzyme present in all living cells responsible for catalysing the reaction, resulting in the blue staining of viable CM-Dil-labelled cells (Figure 2A). Enhanced LDH staining was observed in the in vitro control groups that decreased over time during in vivo treatment in all groups (cells, MNPs +magnet, MNPs −magnet). Quantification of LDH-stained cells revealed a mean 50% loss in cell viability (*p* < 0.001) at day 2 and a mean 90–95% loss across all groups at day 7 compared to the corresponding time point-matched in vitro control, with no influence of MNP labelling nor magnetic activation on cell viability (Figure 2B).

### 3.3. No Adverse Inflammatory Response and No Elevation in Non-Haem Iron Level Detected at Days 2 and 7

CRP (Figure 3A) and non-haem iron (Figure 3B) levels were measured on day 0 (pre-cell implantation) and upon sacrifice on either day 2 or day 7. No deviation from baseline levels (Day 0, pre-implantation levels) for CRP were detected, suggesting an absence of adverse reactions as a result of either the surgery or the presence of the ECM hydrogel/MNPs (Figure 3A). Furthermore, no elevation in non-haem iron levels were detected relative to baseline levels, indicating that the MNPs had not degraded with the release of iron to normal blood circulation levels (Figure 3B).

### 3.4. Cellular Infiltration within the ECM Construct Is Observed for All Groups

The presence of endogenous cellular material, likely to have migrated into the ECM construct, is observed for all groups across both time points (2 and 7 days), as evidenced with H&E staining, where the presence of cells is shown by purple staining (Figure 4). While cells appear evenly distributed at day 2, while evidence of cells accumulating within collagen-rich self-assembled pockets within the ECM is observed at day 7 (Figure 4B, white arrow). No observable difference between treatment groups was recorded. Characterisation of the infiltrated cell population identifies the majority of cells as being CD45-positive, typically a pan leucocyte marker and attributed to the initial inflammatory response (Figure 5A). Although the presence of these cells persisted for the duration of the experiment (7 days), levels appeared markedly reduced at day 7 compared to day 2 (Figure 5A). Furthermore, the presence of endogenous mesenchymal stromal cells was observed across all groups and both time points, as evidenced by STRO-4 staining (Figure 5B). This staining is key in identifying endogenous stromal mesenchymal stromal cells, as the presence of the CM-Dil-labelled cells essentially discounts the implanted or exogenous cell population. The presence of MNPs was further determined by either immunohistochemistry by staining the dextran shell of the particle (Figure 6A) or with Prussian blue stain, which binds to the inner iron oxide core (Figure 6B). MNPs were observed distributed throughout the ECM matrix, and at higher magnifications, co-localisation of the red, anti-dextran stain of the MNP with the orange CM-Dil-labelled oMSCs could be seen. These observations were further validated by Prussian blue staining, which stains the iron oxide core blue. In Figure 6B, intense blue staining was detected within a population of CM-Dil-labelled oMSCs, illustrating that at 7 days post-implantation, therapeutic oMSCs retain the MNP label.

### 3.5. Osteocalcin Expression by Implanted Cells Is Observed in Day 7 ECM Constructs from All Three Groups

Expression of the bone matrix protein osteocalcin was observed in all three groups (cell only, MNPs +magnet, MNPs −magnet) 7 days post-implantation. Co-localisation of osteocalcin with CM-Dil-labelled cells (Figure 7A) suggests that implanted oMSCs are responsible for the production of this protein and indicates they have differentiated towards an osteogenic phenotype. Quantification of osteocalcin expression using at least three images taken from different regions for each sample shows that the total expression was similar across all three groups (Figure 7B). When normalised to the cell number, however, the MNPs +magnet sample demonstrated significantly greater expression than both the cell-only group (*p* < 0.05) and the MNPs −magnet control group (*p* < 0.0001) (Figure 7C).

## 4. Discussion

Creation of a bone defect in any animal model is perceived by the body as an injury, thereby triggering a cascade of fracture repair events. Implantation of a bone tissue-engineered product at this stage following injury will require an interaction between this healing environment and the implanted tissue. The healing process is initiated with an immediate inflammatory response alongside the formation of a haematoma (as a result of blood vessel disruption upon injury) which is essential in enabling bone formation, followed by the recruitment of immune and mesenchymal stromal cells [28,29]. In this study, we report the presence and formation of a haematoma encapsulating the ECM implant in all groups and at both time points, in line with this healing process. At this point, levels of inflammatory mediators, such as the interleukins-1, -6, -11 and -18 (IL-1, IL-6, IL-11, IL-18) and tumour necrosis factor-α (TNF-α), are significantly elevated and effectively work to recruit inflammatory cells [19]. These factors subsequently work to recruit mesenchymal stromal cells and have been known to influence the differentiation and proliferation of stem cells [19,30].

The inflammatory response typically peaks 24 h post-injury and is expected to be completed approximately 7 days later. CD45, a transmembrane glycoprotein, is associated with leukocytes and white blood cells or immune cells, including macrophages and monocytes, which form part of the inflammatory response to injury [5]. Evidence of enhanced infiltration of CD45-positive cells at 2 days post-implantation correlates with these timed events. Interestingly, CD45-positive cells are seen to decline by day 7, irrespective of the treatment group. Our results show a similar trend of CD45 infiltration in the cell-only and both MNP-labelled cells groups. These results suggest that there is no elevated immune response either, as a result of the addition of MNPs to cells or the influence of an external dynamic magnetic field. This was further validated by quantifying C-reactive protein levels, a blood marker of inflammation, in all sheep. In our study, levels remain below 120 µg/mL, a diagnostic requirement for systemic inflammatory responses in sheep. These data provide further evidence for the safety of utilising MNPs in vivo with and without assisted magnetic external fields during therapy. The migration of endogenous mesenchymal stromal cells is also a key event at these early time points [28]. Here, we demonstrate the recruitment of endogenous mesenchymal stromal cells as part of the healing process with the infiltration of STO-4 positive cells (which are not CM-Dil-labelled) observed in all ECM constructs regardless of treatment groups.

Translating regenerative cell-based therapies to pre-clinical studies and beyond is associated with significant hurdles directly impacting therapeutic outcomes. A key element in defining the success in these therapies is refining the dosage of stem cells delivered to the site, which has impacts across multiple clinical indications being treated. Understanding the percentage of stem cells which remain viable post-delivery and are capable of generating new bone tissue to fill the repair site will aid in determining the number of cells required for the therapy [31]. Reports have shown that as much as 99% of cell viability is lost within 24 h of delivery in extreme cardiac therapy applications, whilst other studies reported that “only a few cells” survive at the site of implantation for osteoarthritic stem-cell-based therapies [32,33]. Other pre-clinical studies have further shown that fewer than 5% of injected cells remain at the delivery site within days of implantation [34]. Of course, the animal model, type of injury, stem cell type, mode of delivery and delivery material will affect overall survival and clearance rates. In this pre-clinical model of a critical-sized defect, a significant drop in the survival of donor cells is also observed over 7 days. Interestingly, this is irrespective of treatment group, with no further implications of the MNP label nor magnetic activation, at the timeframes examined, on cellular viability in vivo. Furthermore, we are confident in reporting that within this system, we observe no significant clearance of exogenous, implanted cells. This dramatic loss in cell viability can be attributed to several factors primarily associated with delivering cells within a harsh microenvironment, where nutrient and oxygen supply may be limited and upregulation of chronic inflammatory mediators and increased oxidative stress mechanisms are observed [31,35]. Anoikis is another potential mechanism of cell death in vivo associated with the loss of anchorage-dependent attachment to the ECM [31]. In this study, we suggest that anoikis may not be the cause of cell death, as the viability in in vitro controls is not impaired, which implies that cells are attaching and interacting with the hydrogel effectively. This was further validated in vitro on day 7, where we observed the creation of self-assembled ECM/cell clusters, where cells were beginning to remodel the ECM hydrogel. What is not clear is if there is therapeutic benefit from these delivered cells at the initial stages of therapy in terms of cell signalling, which may enhance migration of endogenous cells to the site of repair. Future work should focus on optimising the gel constitution to determine if changes to ECM can improve the viability of implanted cells.

Mechanotransduction is a highly complex process by which a physical force or series of mechanical stimuli are converted to a set of biochemical signals resulting in controlled cellular responses with a particular emphasis on bone homeostasis and repair [36]. This has spurred the development of novel in vitro technologies to provide mechanical stimuli to effectively trigger mechanotransduction pathways to drive and control osteogenesis in vitro for more efficient production of bone tissue engineering products [37]. We have developed a novel platform technology, where MNPs are used to remotely deliver mechanical stimuli to the mechano-receptor TREK-1, resulting in activation and downstream signalling via an external magnetic array [38]. The MNPs implemented in this study are composed of an iron oxide core and coated with dextran for improved biocompatibility. Overall, MNPs exhibit superparamagnetic properties, suggesting that there is relatively minimal risk (at least over the timeframes examined) of in vivo agglomeration, while enabling efficient manipulation with an external magnetic field. Toxicity and safety are, of course, major concerns in the implementation of MNPs in any biological application. In our study, there is no loss of cell viability, which could be attributed to the MNP label or the application of an external oscillating magnetic field on the MNP-tagged cells within the timeframes examined. Our results agree with other work showing that the use of MNPs in conjunction with stem cells has little or no effect on the proliferation and viability of cells [1,7]. In vitro and ex vivo studies have demonstrated the potential for remote controlled stem cell differentiation [14,39,40,41]. A number of targets including PDGF, TREK-1, RGD and Wnt have been explored as actuators of mechanotransduction pathways in MSCs for bone tissue engineering purposes [39,42,43]. A large animal translational study reported an improvement in bone repair in a critical-sized defect in the medial femoral condyle of a sheep [25]. In this study, we confirm these long-term studies at earlier stages of repair and indicate that further work is needed to study the appropriate cell dosing for therapeutic applications.

## 5. Conclusions

Our study demonstrates how therapeutic cells are substantially reduced in number after implantation in the repair site in all experimental groups. Cell death is accompanied with enhanced leucocyte invasion but not with inflammatory blood marker levels. STRO+ cells are maintained with some levels of osteocalcin expression. MNP labelling, with or without external magnetic fields, does not alter cell number or result in further deleterious impact on cells following implantation.

## Figures and Tables

**Figure 1 cells-10-01776-f001:**
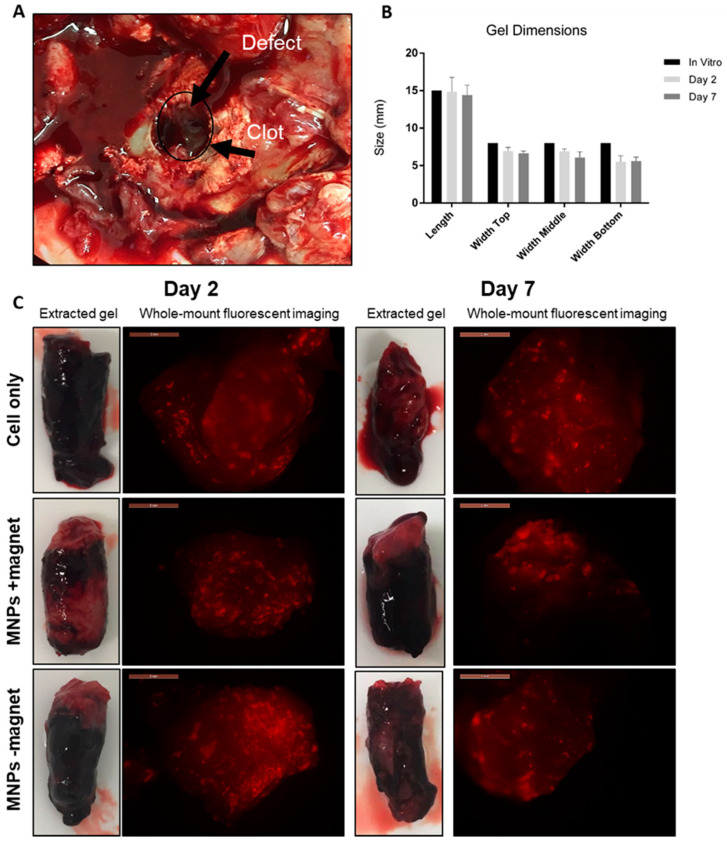
Examination of the implanted construct at 2 and 7 days post-implantation. (**A**) Visualisation of the clot (black circle and arrows) at the site of the defect. (**B**) Quantification of hydrogel dimensions once removed and compared to initial in vitro dimensions. (**C**) Gross evaluation of the removed hydrogel constructs from experimental groups (cell only, MNPs +magnet, MNPs −magnet) with accompanying whole mount fluorescent microscopy with implanted oMSCs are identified in red. Scale = 2 mm.

**Figure 2 cells-10-01776-f002:**
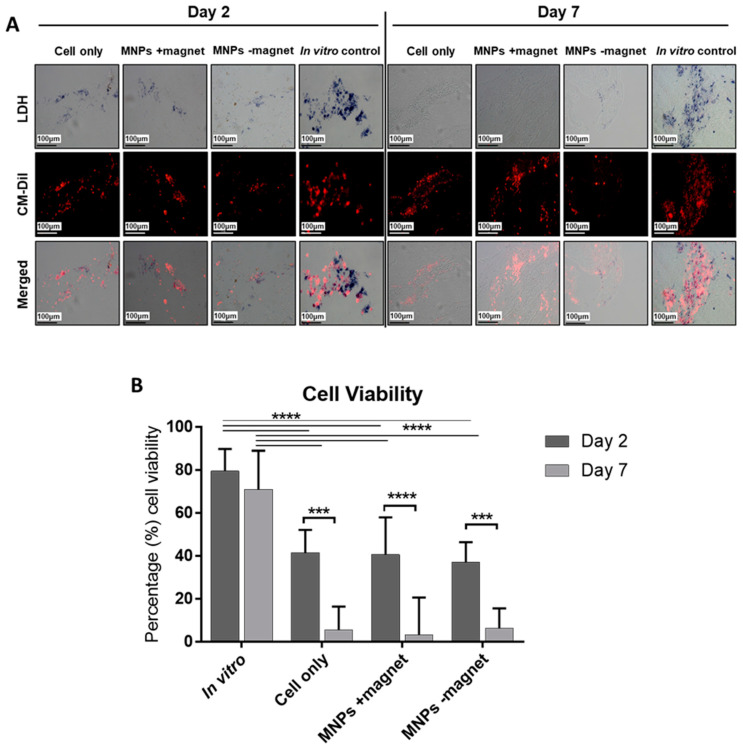
Assessment of oMSC viability at 2 and 7 days post-implantation. (**A**) LDH-stained cryo-sectioned samples of the extracted in vivo construct and time point-matched in vitro controls. Blue staining is indicative of viable oMSCs as determined by the LDH reaction, while red fluorescent staining identifies the CM-Dil-positive implanted oMSCs. The viability of implanted cells was determined by the co-localisation of blue and red fluorescent stains. (**B**) Quantification of cellular viability for all in vivo groups (cells, MNPs +magnet, MNPs −magnet) was undertaken and compared to time point-matched in vitro controls. Data are presented as the average viability (proportion of duel LDH:CM-Dil-labelled cells relative to total CM-Dil-labelled cells) for five random areas per section. Significance was determined by a two-way ANOVA statistical test, where *** is *p* < 0.001 and **** is *p* < 0.0001. Scale bar =100 µm.

**Figure 3 cells-10-01776-f003:**
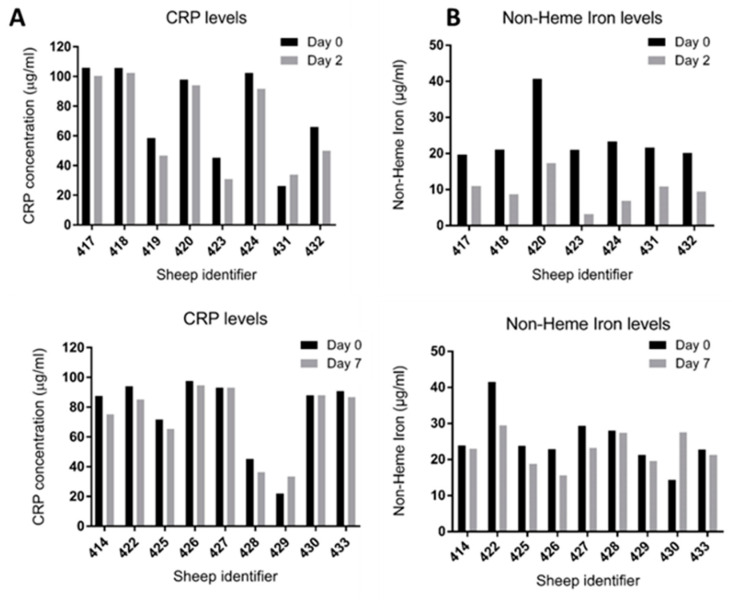
Quantification and comparison of (**A**) the CRP level and (**B**) the non-haem iron levels at day 0 (pre-implantation) and upon sacrifice at either day 2 or day 7. Data are expressed for individual sheep and presented by experimental sheep identifier numbers (414–433).

**Figure 4 cells-10-01776-f004:**
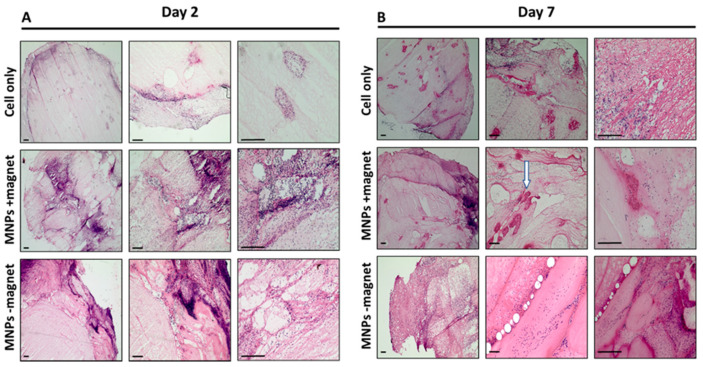
H&E staining of extracted hydrogel constructs at increasing magnification for all groups (cell only, MNPs +magnet, MNPs −magnet) at day 2 (**A**) and day 7 (**B**). Cell nuclei depicted by purple staining and extracellular matrix by pink staining. White arrow showing the pockets of cell accumulation. Scale bar = 100 µm.

**Figure 5 cells-10-01776-f005:**
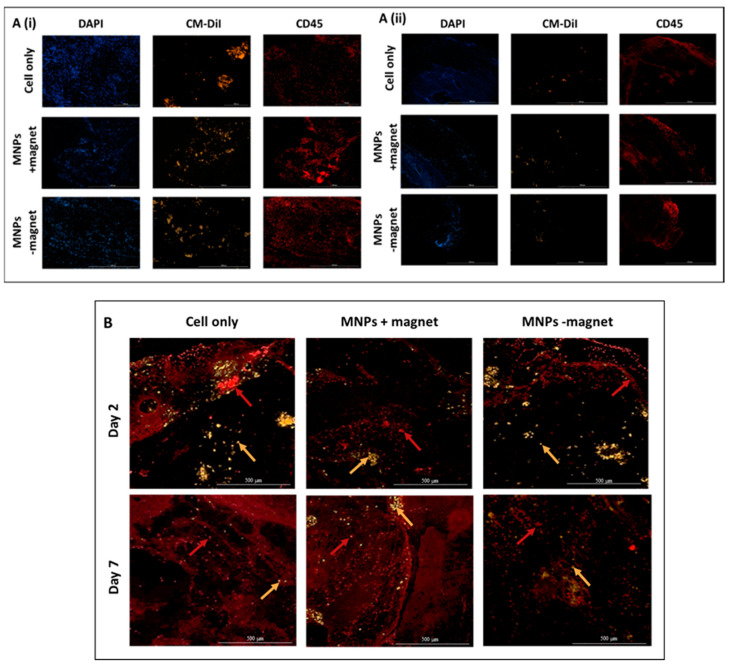
Characterisation of cell infiltration by immunohistochemistry staining for (**A**) CD45 and (**B**) STRO-4 for representative samples at day 2 (Ai) and day 7 (Aii) for the following groups: cell only, MNPs +magnet and MNPs −magnet. DAPI (blue staining) was used for the cell nuclei, CM-Dil (orange fluorescence) identifies implanted oMSCs, whilst the presence of CD45- or STRO-4-positive cells are depicted by red fluorescence. A red arrow indicates an example of STRO-4-positive endogenous cell. An orange arrow indicates an example of the implanted CM-Dil-positive cell. Scale bar = 100 µm (5A) and 5000 µm (5B).

**Figure 6 cells-10-01776-f006:**
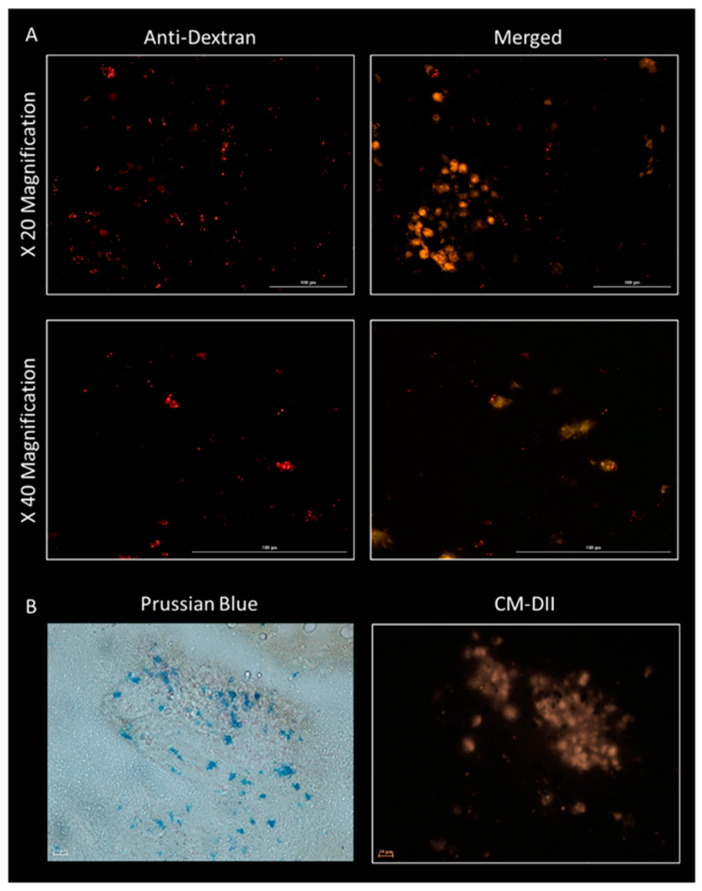
Assessment of MNP degradation in vivo MNPs are composed of an iron oxide core with a dextran coating. (**A**) Immunohistochemistry for dextran (Red Fluorescence). (**B**) Prussian blue staining for the iron oxide core (Blue staining). Implanted oMSCs are identified by the orange membrane-bound stain attributed to the CM-Dil dye. Representative day 7 sample. Scale bar = 100 µm.

**Figure 7 cells-10-01776-f007:**
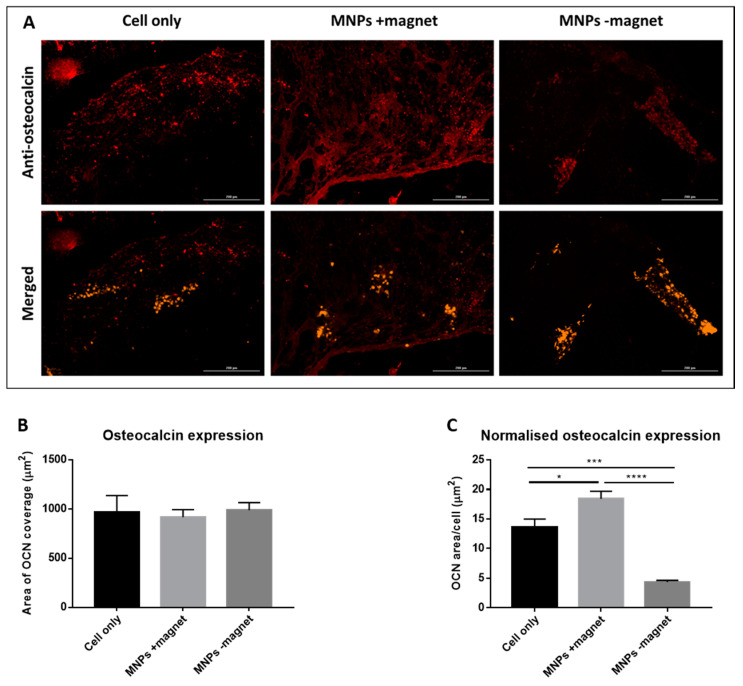
Osteocalcin expression is observed in day 7 extracted hydrogel constructs for all groups (cell only, MNPs +magnet, MNPs −magnet). (**A**) Osteocalcin representative staining at day 7 for all samples. CM-Dil (orange fluorescence) identifies implanted oMSCs, whilst the presence of osteocalcin is depicted by red fluorescence. (**B**) Quantification of osteocalcin expression from images taken from multiple regions of each sample. (**C**) Quantification of osteocalcin expression normalised to cell number, as determined by the number of CM-Dil-positive cells in each image. Significance was determined by a one-way ANOVA statistical test, where * is *p* < 0.05, *** is *p* < 0.001 and is **** is *p* < 0.0001. One sample per group with 3–4 images per sample (n = 3–4). Scale bar = 200 μM.

**Table 1 cells-10-01776-t001:** Details of the experimental groups.

Group	Cells	MNPs	Magnet	CM-Dil Stain	Number of Defects	Time Point
1 (MNPs + magnet)	+	+	+	Yes	6	2 days
2 (Cell only)	+	−	−	Yes	6	2 days
3 (MNPs − magnet)	+	+	−	Yes	6	2 days
4 (MNPs + magnet)	+	+	+	Yes	6	7 days
5 (Cell only)	+	−	−	Yes	6	7 days
6 (MNPs − magnet)	+	+	−	Yes	6	7 days

**Table 2 cells-10-01776-t002:** Details of the antibodies used for immunocytochemistry.

Primary Antibody	Product Code	Secondary Antibody
Anti-Stro-4 (20 µg/mL)	Gift from Professor Andrew Zannettino	Invitrogen, A21236 (10 µg/mL)
Anti-CD45 (10 µg/mL)	WS0544B-100 (Kingfisher Biotech)	Invitrogen, A21236 (10 µg/mL)
Anti-dextran (1 µg/mL)	60026 (Stemcell Technologies)	Invitrogen, A21236 (10 µg/mL)
Anti-osteocalcin	ab13420 (Abcam)	Invitrogen, A21236 (10 µg/mL)

## Data Availability

The datasets generated and/or analysed in the current study are available from the corresponding author on reasonable request.

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
