# Peer review of "Short-Term Evaluation of Cellular Fate in an Ovine Bone Formation Model"

_cells, 2021, doi:10.3390/cells10071776_

Round 1
Reviewer 1 Report
The authors evaluate the viability of STRO+ MSCs in an ovine critical-sized defect model 2 and 7 days post implantation. Moreover, the authors address the incorporation of nanomedicine approaches for multi-model tracking and remote control using external magnet fields.
The manuscript is well written, clear, precise, easy to understand, and offering potentially important information. The following concerns should be addressed before the manuscript can be considered for publication.
- The authors need to explain better why a STRO+ selection of MSCs was performed before implantation.
- Page 4, line 142, please explain what SFM means (serum-free medium).
- In the MNP labelling of STRO+ MSCs section, please explain what is DOTAP.
- Regarding encapsulation of MSCs, what is the composition of the ECM gel? Why did the authors choose the concentration of 12.5 mg/ml?
- Page 4, line 154, please correct “5 x106 MNP…”.
- In the Results section, the authors should provide information about data regarding long-term evaluation of cellular fate, providing information about bone healing from the different groups (MNPs + magnet, Cell only, MNPs - magnet).
- In the Discussion section, the authors should provide some ideas to implement aiming to increase cell viability upon implantation.
Author Response
The authors need to explain better why a STRO+ selection of MSCs was performed before implantation.
Sentence and reference added to section 2.2
Page 4, line 142, please explain what SFM means (serum-free medium). Done
In the MNP labelling of STRO+ MSCs section, please explain what is DOTAP. Done
Regarding encapsulation of MSCs, what is the composition of the ECM gel? Why did the authors choose the concentration of 12.5 mg/ml?
The composition has been added to the text (bovine tibiae).
12.5 mg/ml was chosen as it allowed the formation of pre-formed gels that could then be easily implanted in the defect. Essentially at this concentration, the gels had optimal mechanical properties that allowed their release from the mould and easy insertion in the defect site, as a cylindrical plug. Sentence added to section 2.5.
In the Results section, the authors should provide information about data regarding long-term evaluation of cellular fate, providing information about bone healing from the different groups (MNPs + magnet, Cell only, MNPs - magnet).
Long-term cell fate has been tracked and published in a different paper Markides et al 2018 – this showed how the different groups responded to the treatments.
In the Discussion section, the authors should provide some ideas to implement aiming to increase cell viability upon implantation.
Further work has included looking at gel constitution to determine if changes to ECM effect in vivo viability long-term. Sentence added to discussion (line 391-2).
Reviewer 2 Report
Manuscript ID: cells-1280938
Title: Short-term evaluation of cellular fate in an ovine bone formation model
In this work, the authors aimed to evaluate the short-term (2 and 7 days) implantation of autologous STRO-4 MSCs labelled with a fluorescent lipophilic dye, functionalized with magnetic nanoparticles and encapsulated within a naturally derived bone ECM gel construct, in an ovine critical-sized defect model.
The implanted cells viability, bio-distribution, osteocalcin expression and cellular infiltration within the ECM were analyzed. Moreover, the inflammatory response to the surgery or the presence of ECM hydrogel/MNPs was also investigated.
The study demonstrates a substantial reduction in the number of cells after in vivo implantation, accompanied by enhanced leucocyte invasion but no inflammatory markers were detected. The MNP labelling, with or without external magnetic, does not impact the implanted cell number. Furthermore, STRO-4+ MSCs expressed osteocalcin (7 days post-implantation), indicating that cells have differentiated towards an osteogenic phenotype.
Overall the manuscript is well written, the research design is appropriated to address the main goal of the work, and most results are clearly presented. However, some points might be improved according to the comments below:
Major points:
- In the Materials and Methods section (2.1 Animal experiments) the authors mentioned that nineteen healthy animals were used (line 103). How many animals were used per group? According to Table 1, were used 3 animals per group, which is a total of eighteen. The authors should clarify this point.
- Once the work is based on the transplantation of ovine MSCs, the authors should consider including the cell characterization and phenotyping after isolation (Positive and negative cell markers, differentiation capacity).
- The inflammatory response analysis, by the measurement of CRP and non-heme iron levels, was performed in all animals? In Figure 3 (3.3 No adverse inflammatory response and no elevation in non-heme iron level detected at days 2 and 7), according to the sheep identifier, the CRP (A) and non-heme iron (B) levels were measured in 17 and 16 animals, respectively. Furthermore, why the animal 419 was only subjected to the CRP analysis? The authors should clarify this point and detailed this information in the Material and Methods section.
- According to the Results section (3.4 Cellular infiltration within the ECM construct is observed for all groups), the MNP degradation in vivo was only evaluated at 7 days post-implantation (line 288). Why the assessment of MNP degradation was not evaluated also at 2 days post-implantation? I suggest including the analysis time point in the legend of Figure 6.
- In the Discussion, the authors mention that “here, we demonstrated the recruitment of endogenous mesenchymal stromal cells as part of the healing process with infiltration of STO-4 positive cells (which are not CM-Dil labelled) observed in all ECM constructs regardless of treatment”. However, according to Figure 5B, the STRO-4 staining (red fluorescence) is only evident in the cell-only group at 2 days post-implantation. Can the authors include more representative images that support the presence/recruitment of endogenous mesenchymal stromal cells? I suggest quantifying the STRO-4 staining in different regions of each slide, across all experimental groups.
Minor points:
- I suggest defining the abbreviations oMSCs (ovine MSCs) and SFM (serum-free media) when they first appeared in the manuscript (lines 132 and 151, respectively).
- Authors must use the Celsius degree symbol (°C). Please correct throughout the manuscript.
- I suggest standardizing the way the critical-sized defect is written throughout the manuscript (critical-sized defect/critical sized defect).
Author Response
Minor points 1-3. All amended in the text as suggested.
- In the Materials and Methods section (2.1 Animal experiments) the authors mentioned that nineteen healthy animals were used (line 103). How many animals were used per group? According to Table 1, were used 3 animals per group, which is a total of eighteen. The authors should clarify this point.
This was a typing error. Changed to “eighteen”.
- Once the work is based on the transplantation of ovine MSCs, the authors should consider including the cell characterization and phenotyping after isolation (Positive and negative cell markers, differentiation capacity)
Characterisation and phenotyping has been published in a previous study using the same cell source and isolation/selection techniques (Markides H, McLaren JS, Telling ND, Alom N, Al-Mutheffer EaA, Oreffo ROC, et al. Translation of remote control regenerative technologies for bone repair. npj Regenerative Medicine. 2018;3(1):9, supplementary material). Sentence added to section 2.1.
- The inflammatory response analysis, by the measurement of CRP and non-heme iron levels, was performed in all animals? In Figure 3 (3.3 No adverse inflammatory response and no elevation in non-heme iron level detected at days 2 and 7), according to the sheep identifier, the CRP (A) and non-heme iron (B) levels were measured in 17 and 16 animals, respectively. Furthermore, why the animal 419 was only subjected to the CRP analysis? The authors should clarify this point and detailed this information in the Material and Methods section.
CRP done on only 17 animals because one sample was damaged during preparation and could not be used.
Non—heme iron levels done on only 16 animals because two samples were damaged/lost during preparation and could not be used.
Non-heme iron levels not included for sheep 419 because that sample was lost during preparation of the non-heme iron assay, but the sample reserved for CRP analysis was intact.
The numbers reflect the number of viable samples we were able to analyse in the experimental protocol.
- According to the Results section (3.4 Cellular infiltration within the ECM construct is observed for all groups), the MNP degradation in vivo was only evaluated at 7 days post-implantation (line 288). Why the assessment of MNP degradation was not evaluated also at 2 days post-implantation? I suggest including the analysis time point in the legend of Figure 6.
Assessment of MNP degradation was evaluated at both time points, but we have included one representative day 7 image to demonstrate this point – if the iron oxide core is still present at day 7 then it follows that it was also present at day 2. We have added the time point to the image for clarification.
- In the Discussion, the authors mention that “here, we demonstrated the recruitment of endogenous mesenchymal stromal cells as part of the healing process with infiltration of STO-4 positive cells (which are not CM-Dil labelled) observed in all ECM constructs regardless of treatment”. However, according to Figure 5B, the STRO-4 staining (red fluorescence) is only evident in the cell-only group at 2 days post-implantation. Can the authors include more representative images that support the presence/recruitment of endogenous mesenchymal stromal cells? I suggest quantifying the STRO-4 staining in different regions of each slide, across all experimental groups.
We have improved the images in Figure 5 and added more arrows, which we hope, will better illustrate the point that endogenous red-stained cells are present in all groups. Our aim is to indicate that there has been endogenous cell recruitment in each group, not that one group encourages more or less recruitment, which would require further animal experiments. Quantification of cell infiltration in these sections is more challenging; the conclusion would not be accurate as there are numerous cell clusters that are difficult to accurately count using fluorescent imaging.
Reviewer 3 Report
The manuscript entitled: „Short-term evaluation of cellular fate in an ovine bone formation model” by Markides et al is of high interest for the readers of CELLS.
The authors investigate the number and survival of transplanted autologous ovarian MSC in an ovarian critical size defect model over an observation period of 7 days. A significant decrease in implanted MSC at day 7 is observed. Another aspect that was investigated is whether additional loading of the transplanted cells with magnetic nanoparticles (MNP) and subsequent magnetic stimulation lead to differences in cellular survival. However, this could not be clearly confirmed.
A very important point is addressed in this work, still very little is known about the fate of MSC implanted into a bone defect in association with a bone graft substitute or extracellular matrix. In agreement with the few existing works, a significant decrease of MSC in implanted ECM constructs could be measured. An important aspect represents the identification of transplanted cells. Here, prestaining with CM-DiL was performed. This is an adequate approach for cell tracking with relatively short observation periods. Furthermore, by detection of LDH activity, it was possible to distinguish histologically between viable and non-viable cells.
Overall, a methodologically elaborate study is presented, the manuscript has generally been written in a very comprehensible manner, yet a few points still need to be addressed.
- Materials and Methods: Some more information about the ECM is needed, what are its main constituents?
- Does the staining with DiL-CM (and especially) the solvent DMSO harm the cells? It is my (unpublished) observation that other cell tracking dyes like CFSE affect cellular activity. This may be added to the discussion.
- Materials and Methods: Provide source for ImageJ
- Materials and Methods: Define „DOTAP“
- Materials and Methods: Please provide an extra statistics paragraph. Microscopic analyses, were the technical replicates per specimen averaged and these averages used for statistical analysis? Please clarify.
- Results: Figure 3B: Legend: „Day 2” should be “Day 7”. Furthermore it seems that the difference between D2 and D7 is significant, all animals demonstrate higher values on day 2 (at least minimum 100% higher compared to day 7 in every animal). However, the authors expected an increase from D2 to D7, which could be induced by the degradation of MNP. How can the, in my opinion, significant decrease of the non-Heme-Iron-level be explained?
- Results: Figure 5 A: Based on the microscopic image it seems that leukocyte infiltration is more pronounced in MPN+mag-group compared to other groups. Were the CD45+ cells counted and were the groups statistically compared? If possible, these data should still be integrated and discussed in the manuscript.
- Discussion: How specific is STRO-4 for ovine MSC, is it also expressed on other cell types? Please provide some more information.
- Formal aspect: Check superscript digits and characters throughout the document.
- Just my personal interest: Probably it’s a completely stupid question: I wonder, if the magnetic beads used for MSC enrichment still stuck to the cells and thus could be also stimulated somehow by the external magnetic field after implantation (That group did not receive magnetical stimulation in your study). Do you have any experience on this?
Author Response
- Materials and Methods: Some more information about the ECM is needed, what are its main constituents? Done
- Does the staining with DiL-CM (and especially) the solvent DMSO harm the cells? It is my (unpublished) observation that other cell tracking dyes like CFSE affect cellular activity. This may be added to the discussion.
We have never observed any loss in cell viability following introduction of CM-Dil dye. This was tested extensively in numerous cell types when we first started using the reagent in our lab, but that data is not relevant to this study. It is widely used in our own labs and others. CM-Dil was chosen specifically because it is known to have low cytotoxicity.
https://www.thermofisher.com/order/catalog/product/C7000#/C7000
Below is an example of a study on MSCs which supports this (figure 3):
https://journals.plos.org/plosone/article?id=10.1371/journal.pone.0134920
- Materials and Methods: Provide source for ImageJ Done
- Materials and Methods: Define "DOTAP“ Done
- Materials and Methods: Please provide an extra statistics paragraph. Microscopic analyses, were the technical replicates per specimen averaged and these averages used for statistical analysis? Please clarify.
Section added (2.10).
Microscopic analyses: owing to limited numbers of samples, only one specimen for each group was sectioned and stained. Multiple images were taken from different regions of the same section and these were used to calculate osteocalcin expression (as stated in figure 7 legend).
- Results: Figure 3B: Legend: "Day 2” should be “Day 7”.
We have double-checked the figure legend for 3B and cannot see any error.
Furthermore it seems that the difference between D2 and D7 is significant, all animals demonstrate higher values on day 2 (at least minimum 100% higher compared to day 7 in every animal). However, the authors expected an increase from D2 to D7, which could be induced by the degradation of MNP. How can the, in my opinion, significant decrease of the non-Heme-Iron-level be explained?
The point of doing the non-heme iron assay was to show that the particles had not degraded and entered the blood stream. If they had there would be a sharp increase, at either time point. Why they are lower at day 2 than day 7 is probably linked to their diet, as the time of feeding is fairly consistent. It was not deemed a concern by the veterinarians assisting in the study.
- Results: Figure 5 A: Based on the microscopic image it seems that leukocyte infiltration is more pronounced in MPN+mag-group compared to other groups. Were the CD45+ cells counted and were the groups statistically compared? If possible, these data should still be integrated and discussed in the manuscript.
It was not possible to accurately count CD45+ cells from our images, as numerous pockets of high cell density made it impossible to distinguish individual cells in many areas. High levels of background fluorescence from the ECM construct also meant that fluorescence intensity could not be used to estimate levels of CD45+ cells.
- Discussion: How specific is STRO-4 for ovine MSC, is it also expressed on other cell types? Please provide some more information.
This has been addressed in relation to reviewer 1 comments and a reference added to the manuscript along with an explanation (section 2.2).
- Formal aspect: Check superscript digits and characters throughout the document. Done
- Just my personal interest: Probably it’s a completely stupid question: I wonder, if the magnetic beads used for MSC enrichment still stuck to the cells and thus could be also stimulated somehow by the external magnetic field after implantation (That group did not receive magnetical stimulation in your study). Do you have any experience on this?
Magnetic beads used for cell sorting are not suitable for this application, however, tagging CD markers used for MSC enrichment with other types of particles could be used for downstream magnetic activation and is the subject of further study in our laboratory.